# Bayesian Sequential Design for Identifying and Ranking Effective Patient Subgroups in Precision Medicine in the Case of Counting Outcome Data with Inflated Zeros

**DOI:** 10.3390/jpm13111560

**Published:** 2023-10-30

**Authors:** Valentin Vinnat, Djillali Annane, Sylvie Chevret

**Affiliations:** 1ECSTRRA Team, INSERM U1153, Université Paris Cité, 75010 Paris, France; sylvie.chevret@u-paris.fr; 2Intensive Care Unit, Raymond Poincaré Hospital, 78266 Garches, France; djillali.annane@aphp.fr; 3Institut Universitaire de France (IUF), 75231 Paris, France

**Keywords:** personalized medicine, biomarkers, Bayesian inference, identification of subset

## Abstract

Precision medicine is revolutionizing health care, particularly by addressing patient variability due to different biological profiles. As traditional treatments may not always be appropriate for certain patient subsets, the rise of biomarker-stratified clinical trials has driven the need for innovative methods. We introduced a Bayesian sequential scheme to evaluate therapeutic interventions in an intensive care unit setting, focusing on complex endpoints characterized by an excess of zeros and right truncation. By using a zero-inflated truncated Poisson model, we efficiently addressed this data complexity. The posterior distribution of rankings and the surface under the cumulative ranking curve (SUCRA) approach provided a comprehensive ranking of the subgroups studied. Different subsets of subgroups were evaluated depending on the availability of biomarker data. Interim analyses, accounting for early stopping for efficacy, were an integral aspect of our design. The simulation study demonstrated a high proportion of correct identification of the subgroup which is the most predictive of the treatment effect, as well as satisfactory false positive and true positive rates. As the role of personalized medicine grows, especially in the intensive care setting, it is critical to have designs that can manage complicated endpoints and that can control for decision error. Our method seems promising in this challenging context.

## 1. Introduction

The main aim of controlled clinical trials is to evaluate the efficacy of an experimental treatment over a control. However, there has been a growing interest in precision medicine, a new paradigm motivated by the possibility that patient responses to a particular treatment are heterogeneous, which may be due to patient biological profiles [1]. Even though the one-size-fits-all strategy helps establish a new standard of care for the general population, the identified treatment might not be the best option for some subsets of patients. Several statistical methods and clinical trial designs have been proposed to democratize the precision medicine approach, including biomarker-stratified clinical trial designs [2,3,4,5], as reviewed by Simon [6]. Optimal individualized treatment rules built on biomarker further aim to identify a subgroup of patients who benefit from the experimental treatment and aid in determining personalized treatment decisions [7,8,9,10,11]. Likewise, fixed or adaptive enrichment designs use biomarkers to restrict enrollment to patients who are expected to benefit more from the experimental treatment than the control, which magnifies the signal and improves the power to detect the treatment effect [12,13,14,15,16].

Researchers have recently proposed frequentist approaches to identifying subgroups of interest. Lipkovich et al. [17] developed a frequentist nonparametric recursive partitioning method to analyze subgroup treatment effects. Another nonparametric method, random forests of interaction trees (RFIT), was proposed by Su et al. [18] to estimate subgroup treatment effects. Additionally, Foster et al. [19] created the virtual twins’ method, and Altstein et al. [20] suggested a new computational method for parameter estimation of an accelerated failure time (AFT) model with subgroups identified by a latent variable. Bayesian designs have potential benefits over frequentist designs for prospective personalized randomized controlled trials (RCTs) since they naturally extend from simple [21] to more complex but efficient models [22] while being highly flexible and thus facilitating early decision-making via planned or unplanned interim analyses [23]. Bayesian inference also provides the probability that a treatment is the best for a particular subgroup [24]; such probabilistic statements have a straightforward interpretation and are thus friendly to scientific researchers with some statistical background. Bayesian adaptive designs are naturally highly flexible and allow for direct probability computations at any trial point while accounting for the uncertainty in the parameters of interest. Another advantage of Bayesian analyses is the integration of prior knowledge about the treatment effect in each subgroup. Finally, Bayesian adaptive designs can illustrate the effectiveness of a treatment in the overall population or subpopulations with higher power when compared to that of a fixed design of the same size [25].

This research focuses on a prospective study design aiming at defining which patient subgroup benefits from the treatment among several subsets that have already been identified. Most proposed approaches used a binary or a continuous outcome measure, while we focused on counting data with inflated zeros. Such data are frequently used in critical care clinical studies of days without organ failure, including vasopressors or mechanical ventilation [26,27,28]. They are commonly reported composite outcome measures in randomized clinical trials conducted in intensive care patients, as ways of quantifying their survival while accounting to their severity status, by defining “failure-free day” composite outcomes. These outcomes are often used as primary or secondary outcomes in RCTs or observational studies of critically ill patients, such as patients with sepsis shock [26] or COVID-19 [29]. For instance, the vasopressor-free days in sepsis patients combines survival and duration of vasopressors in a manner that summarizes the “net effect” of the treatment on these two outcomes. Their values are usually nonnegative and have excessive zeros and dispersion. Indeed, an excess of zero event-free days is usually observed due to the high proportion of deaths in intensive care units (ICUs). In fact, a usual practice in ICU studies is to assign zero to the count response outcome when the patient died before a follow-up is completed [27]. Several statistical models have been proposed for analysis to address these challenges, including zero-inflated models [30], two-part models, and beta-binomial models [31]. A data truncation over the 28 first days is often performed, due to the common length of stay in ICU of those patients, which should be further handled using a Zero-Inflated Truncated model.

We developed a Bayesian sequential clinical trial design that evaluates the therapeutic intervention and identifies the subset of patients who respond better to the experimental therapy based on the surface under the cumulative ranking curve (SUCRA) method [32]. Although this method has been developed for network meta-analyses in order to rank different drugs based on their estimated effects, it is useful for ranking subgroups of patients in our context and thus for identifying the best responding and predictive subgroup of interest. In each subpopulation and overall, the count outcome was modeled by a Bayesian zero-inflated truncated Poisson model. The SUCRA method was then applied to the posterior ranking distribution of subgroups according to their treatment effect.

The rest of this paper is organized as follows. In Section 2, we describe a motivating trial for intensive care units and propose a design structure, probability model, and methods to identify subsets in which the treatment is most effective. In Section 3 and Section 4, we evaluate the operating characteristics of the proposed design by using simulation studies. We provide a discussion and conclusion in Section 5 and Section 6.

## 2. Method

### 2.1. Motivating Example

Rapid recognition of corticosteroid resistance or sensitive sepsis (RECORDS) is an ongoing phase III trial study that started in February 2020 and should be completed by October 2024 (NCT04280497) [33]. This is a multicenter pragmatic double-blinded randomized controlled trial with broad eligibility criteria that include all patients admitted to the ICU with a primary diagnosis of sepsis. Patients are randomly assigned to hydrocortisone with the addition of fludrocortisone or placebo for seven days, targeting 1800 patients with complete follow-ups up to six months.

A sequential design was used that evaluates the therapeutic intervention of targeted therapy and identifies among several predefined subsets those responding the best to the experimental therapy. Sequential analyses use the number of vasopressor-free days out of 28 days as the measure of efficacy and the occurrence of severe adverse events within the first 28 days as the measure of toxicity.

### 2.2. Design Structure

Motivated by the RECORDS study, we considered a group sequential clinical trial with *K* analyses. Patients are individually randomized to experimental or control treatments. Our design sequentially enrolls a maximum of *N* patients by cohorts of size $n_{1},\ldots ,n_{K}$ with $N=\sum_{k=1}^{K}n_{k}$.

Let us consider a set of *m* biomarkers of interest that may or may not be measured for each patient at study entry. For patient *i*, $X_{mi}=\left[0,1,NA\right]$ is the measurement of biomarker *m*, where 0 and 1 denote the absence and presence of the characteristic *m*, respectively, whereas NA denotes a missing value. The biomarker, denoted as $X_{m}$, divides the sample into two subgroups of patients. These subgroups can overlap, as a patient may have multiple biomarkers measured. To define subsets of patients based on biomarker measurements, we can partition the set of all enrolled patients Ω into *J* partitions. Let $S_{j}$ denote the subset of patients corresponding to a specific combination of biomarker values, and it may contain patients for whom some were not measured. We have $J=3^{m}-1$ possible partitions, each corresponding to a distinct combination of biomarker values, and the partition corresponding to none of the biomarkers being measured is empty. By analyzing treatment effects for each subset separately, we can investigate whether treatment efficacy varies across patient subgroups based on the available biomarker measurements.

The trial begins by enrolling patients, whichever their subset, up to the interim sample size of $n_{1}$ patients randomly allocated to experimental or control treatments with equal probabilities. For each interim analysis k=1,…,K−1, performed when $n_{k}$ patients have been enrolled and their outcomes are available, the superiority of the experimental treatment against the control is evaluated marginally, and for each subset, by using all accumulated data. The subsets are then ranked from the one benefiting the most from the experimental treatment to the one for which it is the least likely to be efficient. If there is some evidence that the experiment is superior to control for the global trial population, the trial is terminated with the conclusion of the benefit of experimental treatment in the whole population. Otherwise, if the maximum sample size *N* is not reached, the next cohort of patients $n_{k+1}$ is recruited. If the trial is not stopped early at an interim analysis, a final analysis is performed with the maximum sample size *N* when the last patient outcome has been evaluated.

### 2.3. Probability Model

For each patient *i*, let $Y_{i}$ be the number of event-free days out of *G*. Thus, the outcome is continuous with an upper bound of *G* and depends on the biomarker statuses, the treatment, and the interaction between the treatment and the biomarker profiles.

We modeled the outcome distribution, $Y_{i}$, $0\leq Y_{i}\leq G$, by using the zero-inflated truncated Poisson (ZITP) regression model proposed by Tsai and Lin [34], with *G* as the truncation parameter, where G>0. The ZITP model assumes that two processes generate the count data. The first process is the truncated Poisson distribution, and the second is a process that generates the excess zeros, that is, responses of zeros that cannot be explained by the truncated Poisson distribution, with a zero-inflation rate $\phi_{i}$. The ZITP model derived from the zero-inflated Poisson (ZIP) regression model described by Lambert [30] can be written as follows:(1)$$\begin{array}{cc} Y_{i}{∼}_{i=1,\ldots ,n}^{indep.}P\left(Y_{i}=y_{i};\lambda_{i},\phi_{i}\right) & =\phi_{i}I_{\left\{0\right\}}\left(y_{i}\right)+\left(1-\phi_{i}\right)f\left(y_{i}\right)\\ \; & =\left\{\begin{array}{cc} \phi_{i}+\left(1-\phi_{i}\right)f\left(0\right) & ,y_{i}=0\\ \left(1-\phi_{i}\right)f\left(y_{i}\right) & ,y_{i}\in \left[1,G\right] \end{array}\right. \end{array}$$
where *f* is the truncated Poisson density:(2)$$f\left(y_{i}\right)=\left(\frac{\lambda_{i}^{y_{i}}}{y_{i}!}\right){\left(\sum_{z=o}^{G}\frac{\lambda_{i}^{z}}{z!}\right)}^{-1}I_{\left\{0,\ldots ,G\right\}}\left(y_{i}\right),$$
with $\lambda_{i}$, the mean of the standard Poisson distribution, which can be modeled as a linear combination of the covariates:(3)$$\log\left(\lambda_{i}\right)=\beta_{0}+\sum_{j=1}^{m}\beta_{X_{j}}X_{ji}+\beta_{T}T_{i}+\sum_{j=1}^{m}\beta_{inter_{j}}X_{ji}T_{i}$$
where $\beta_{0}$ is the baseline log number of event-free days, $X_{i}$ is the vector of *m* biomarkers, $T_{i}$ is the treatment indicator, with $\beta_{X}$, $\beta_{T}$, their associated vector of coefficients, respectively, and $\beta_{inter}$ represents the interactions between the treatment and the biomarkers.

The probability that the outcome of patient *i* is generated by the excess zero mechanism $\phi_{i}$ is assumed to be independent of the treatment, the biomarker statuses, and their interactions (that is, $\phi_{i}=\phi$), although one can model it according to some chosen covariates. Thus, the zero-inflation rate ϕ was defined from a logistic regression model as follows:(4)$$logit\left(\frac{\phi }{1-\phi }\right)=\gamma_{0}$$

Here, $\gamma_{0}$ is the baseline log odds of excess zeros.

### 2.4. Decisions Rules

Early stopping rules can be incorporated based on the probability of relevant clinical events (e.g., [35]), for instance, if there is sufficient information to declare that one treatment is more efficacious than the other. Let $\lambda_{T}$ be the ZITP λ parameters estimated from the whole sample in the treatment arm *T*, where T=1 denotes the experimental arm and T=0 denotes the control arm. For such estimations, we placed ourselves in a Bayesian framework, using noninformative priors with large variances to let most of the information arise from the data. This further allowed us to define probabilistic statements regarding treatment effects overall and in subsets from which stopping rules were derived. We considered the efficacy stopping rule from Thall and Simon [35], thus stopping the trial for efficacy if there is enough evidence of a meaningful difference in efficacy in favor of the experimental treatment, based on the posterior probability $Pr(\lambda_{T=1}-\lambda_{T=0}>\delta |Data)$. The clinically relevant threshold δ and the decision threshold ϵ were optimized based on a desired false-positive rate close to 5–10% through a grid search with the maximum sample size.

Action triggers for decision-making were derived from Harrell [36], Ohwada [37], and Morita [38]:If $Pr(\lambda_{T=1}-\lambda_{T=0}>\delta |Data)>\epsilon$, the experimental treatment is reported to be superior and stop the trial.If the trial has not yet reached its maximum sample size, it continues.

## 3. Simulation Study

The statistical performances of the proposed design were evaluated through a series of simulations.

### 3.1. Data Generation

Our simulations involved a maximum of 1800 patients, equally randomized to receive either the control or the experimental treatment denoted as $T_{i}=\left\{0,1\right\}$, where $T_{i}=1$ denotes the experimental group.

We considered two binary biomarkers, $x_{1}$ and $x_{2}$, which take the value 0 (negative) or 1 (positive) and are independently generated from a Bernoulli distribution B(0.5). To emulate real-world scenarios in clinical trials where not all biomarkers are always measured, we introduced a missing data mechanism that is missing completely at random (MCAR). Specifically, we randomly distinguished three equal-sized groups (terciles), one group with both $x_{1}$ and $x_{2}$ measured, a second group with only $x_{1}$ measured, and a third group with only $x_{2}$ measured. This approach allowed us to create a realistic dataset reflecting the different scenarios of biomarker measurements in clinical trials. Given the different possible states of the biomarkers (positive, negative, or missing) and considering that at least one biomarker is measured in each individual, the study population may be divided into $3^{2}-1=8$ different subsets based on the $x_{1}$ and $x_{2}$ values, allowing a detailed analysis of the different patient subgroups. These subsets were further categorized into “unmissed” and “missed” groups to enhance the analysis. The “unmissed” groups represent the four combinations of the biomarkers when both are measured (i.e., ++, +−, −+ and −−), while the “missed” groups refer to the four scenarios where only one biomarker is measured (i.e., +?, −?, ?+, and ?−).

Responses $Y_{i}$ were generated from a zero-inflated truncated Poisson model. To this end, for each patient *i*, we first generated a random variable $W_{i}$ from a traditional Poisson ($\lambda_{i}$) distribution. Then, let $R_{i}$ be a random variable following a truncated Poisson distribution with an upper bound *G* computed for each patient as $R_{i}=min\left\{W_{i},G\right\}$. For each patient *i*, $W_{i}$ was generated from a Poisson distribution with parameter $\lambda_{i}$ computed with Equation (Equation 5). Finally, to mimic zero inflation in the motivating example data, we generated a proportion ϕ of zero inflation with an auxiliary random variable $U_{i}∼$ Uniform [0, 1]. If 0 < $U_{i}$ < ϕ, then $Y_{i}=0$; if ϕ < $U_{i}$ < 1, then $Y_{i}=R_{i}$. ϕ was fixed at 30% for all scenarios, as observed in previous studies in sepsis [39].
(5)$$log\left(\lambda_{i},x_{1i},x_{2i},T_{i}\right)=\beta_{0}+\beta_{1}x_{1i}+\beta_{2}x_{2i}+\beta_{T}T_{i}+\beta_{inter_{1}}T_{i}x_{1i}+\beta_{inter_{2}}T_{i}x_{2i}$$
with $\beta_{1}$ and $\beta_{inter_{1}}$ being set to 0 if $x_{1}$ is missing and $\beta_{2}$ and $\beta_{inter_{2}}$ being set to 0 if $x_{2}$ is missing.

### 3.2. Data Analysis

One interim analysis was planned to occur when the 500th patient outcome was recorded, with the final analysis performed after the last patient was enrolled and their outcome was measured.

For each analysis, we considered only individuals with non-missing biomarker data, either both $x_{1}$ and $x_{2}$, only $x_{1}$ and only $x_{2}$, separately. Zero-inflated Truncated Poisson models were fitted. In the case of all the patients having measurements of both biomarkers, the model included both $x_{1}$ and $x_{2}$ as well as the interaction term. In the case of missing data on either biomarker, two situations were considered: either estimating the treatment effect in patients with measures of $x_{1}$, then of $x_{2}$, separately, or estimating the treatment effect once in the subset of patients with both measures of $x_{1}$ and $x_{2}$. In each of these three analyses, the difference in the treatment effect was estimated, based on a Bayesian approach, with noninformative normal priors with mean 0 and large standard deviation (10${}^{4}$) for all regression coefficients. Posterior distributions were computed by using a Markov chain Monte Carlo (MCMC) sampling method. Three chains were implemented, with an initial burn-in of 10,000 samples followed by an additional 30,000 samples retained for computing posterior distributions.

Following on, the overall treatment effect was derived by computing a weighted average of these three estimates, weighted by the observed prevalence of each case. The stopping rule described above was then computed at the interim analysis.

Last, at the time of decision to stop (either at the time of interim analysis if the stopping rule was fulfilled or at the time of terminal analysis), we applied the surface under the cumulative ranking curve (SUCRA) approach, proposed by Salanti et al. [32]. It is a Bayesian approach that ranks treatment groups from a network meta-analysis based on their estimated measure of efficacy. Instead of ranking treatments, we applied SUCRA to rank the subgroups defined by their biomarker values based on their predictive treatment effects.

We first considered all patients by ranking the subsets according to each biomarker $x_{1}$ or $x_{2}$ separately (whichever the other was measured or not). Next, we only considered the complete subset of patients who had both measurements (that is, one-third of the whole sample).

### 3.3. Scenarios

We assessed the performance of the design under several distinct scenarios. These scenarios were chosen to represent a range of potential real-world situations, from no association between the biomarker and outcome to complex interactions, as described in Table 1.

Scenario 1 represents the null scenario with no prognostic value of biomarker and no treatment-by-subset interaction. Scenario 2 indicates a prognostic value of biomarker $x_{1}$ ($\beta_{1}\neq 0$). Scenario 3 demonstrates a predictive value of biomarker $x_{2}$ ($\beta_{inter_{2}}\neq 0$). In Scenario 4, both biomarkers, $x_{1}$ and $x_{2}$, show similar prognostic values and significant interactions with the treatment. This suggests that the treatment is more effective in patients with positivity for either biomarker $x_{1}$ or biomarker $x_{2}$, as indicated by nonzero coefficients ($\beta_{inter_{2}}\neq 0$ and $\beta_{inter_{1}}\neq 0$).

In scenario 5, both biomarkers have prognostic values, with a clear qualitative treatment-by-subset interaction observed for biomarkers $x_{1}$: Treatment proves detrimental for patients testing positive for biomarkers $x_{1}$ but proves beneficial for those testing negative. A quantitative treatment-by-subset interaction is observed for biomarkers $x_{2}$ ($\beta_{inter_{2}}>0$ and $\beta_{inter_{1}}>0$). Comparatively, scenario 6 mirrors the conditions and outcomes observed in scenario 5 but shows an increased treatment effect.

### 3.4. Outputs

In the study, we investigated the operating characteristics of the design in each trial through 1000 independent replications. The clinically relevant threshold δ was set at a difference of 2 days without vasopressors out of 28 days. To control the false-positive rate (that is, the proportion of simulations concluding in efficacy under the null Scenario 1) between 5% and 10%, the decision threshold ϵ was set to 0.995 after a grid search.

Among the 1000 replications, to consider the performances of the proposed approach as satisfactory, we computed the false positive rate and the true positive rate related to treatment effect, regardless of the biomarker statuses under the null scenario or alternative, respectively. We first considered the correct decisions regarding the treatment effect in the whole sample, as measured by the type I error of the test (defined as the rate of false positive conclusions to a treatment effect under the null) and the power (defined as the rate of true positive decisions of a treatment effect under the alternative). We then considered the availability of the design to detect the subset of patients (as defined by their biomarker values) who benefit the best from the treatment. This was measured on the proportion of correct identification of the most predictive biomarker subset. We also reported the early stopping rate with a conclusion of treatment efficacy and the average sample size. Following that, based on the SUCRA obtained from each sample, we computed the distribution of ranks of each subset.

Sensitivity analyses were finally conducted to assess the robustness of the approach regarding the prevalence of $x_{2}$ positivity (while that of $x_{1}$ was set to 0.5). Five prevalence rates for the positivity of the biomarker $x_{2}$ (0.2, 0.4, 0.5, 0.6, 0.8) were used in those patients with available measurements of $x_{2}$.

All analyses were performed by using R version 4.0.1 [40]. The R2jags package was used for MCMC [41]. All codes are available upon request.

## 4. Results

Table 2 reports the results of the approach regarding the overall treatment effect at the time of treatment stopping in the different scenarios, for varying values of biomarker positivity $x_{2}$.

In null scenario 1, regardless of the prevalence of the positive biomarker $x_{2}$, the false-positive rate was below 10% with an average sample size of approximately 1730 patients and an early stopping rate of approximately 5%, as expected. In scenario 2, the true positive rate increased to approximately 20% regardless of the prevalence of the positive biomarker $x_{2}$. Indeed, the true positive rate did not vary between the different prevalence of the positive biomarker $x_{2}$ as there is only an effect of the positive biomarker $x_{1}$ and therefore had no impact on the results.

In both scenarios 3 and 4, the true positive rate was affected by the prevalence of biomarkers $x_{2}$. In scenario 3, due to the significant predictive power of the $x_{2}$ biomarker, the true positive rate increased from 0.437 to 0.999 as the prevalence of this biomarker increased from 0.2 to 0.8, respectively. In scenario 4, even at a lower prevalence of 0.2 of $x_{2}$, the true positive rate was remarkably high at 0.888, underscoring the combined power of both prognostic and predictive effects in determining outcomes. The average sample size decreased over time in these scenarios, which is consistent with the increased early discontinuation rate.

In Scenario 5, due to a pronounced qualitative interaction against biomarker $x_{1}$, it consistently pushes the overall treatment effect below the predetermined clinical threshold of 2. This effectively suppresses the potential positive effect of treatment across varying prevalence of biomarkers $x_{2}$. Consequently, regardless of the prevalence dynamics of biomarkers $x_{2}$, the true positive rate remains stubbornly close to zero, suggesting that the treatment effect is significantly influenced by the negative interaction with biomarkers $x_{1}$.

In scenario 6, an increase in the prevalence of biomarker $x_{2}$ leads to a rise in the true positive rate, shifting from 0.015 to 0.334 with a prevalence of 0.2 and 0.8, respectively. This was parallel to a growth in the early stopping rate, ranging from 0.011 to 0.137, indicating quicker trial conclusions with higher prevalence. Moreover, the average sample size decreased as the prevalence of biomarkers $x_{2}$ escalated, revealing heightened trial efficiency under these conditions. Thus, scenario 6 highlights the positive impact of higher biomarker $x_{2}$ prevalence on treatment success and trial efficacy.

Figure 1 displays the delineation of the predictive efficacy observed across different subgroups, using the SUCRA approach. It stacks for each subset the estimated probability of being ranked at the 1st, 2nd, 3rd, and 4th place by the SUCRA approach, in several situations regarding the availability of both biomarker values. Actually, in Figure 1a, the four patient subgroups were formed by only taking into account one biomarker, separately. Conversely, the ranking probabilities displayed in Figure 1b were computed on the subset of patients for which both biomarkers were measured. Thus, the four patient subgroups were formed by taking into account the status of both biomarkers simultaneously.

Figure 1a shows a uniform distribution of rankings among subgroups in Scenario 1, where each subgroup has an approximately 25% chance of achieving any given rank. This distribution highlights the marginal influence of the biomarker in this context. In Scenario 2, while there is a slight prognostic effect toward biomarker $x_{1}$, the distribution largely resembles that of Scenario 1. Scenario 3 highlights the dominance of the positive $x_{2}$ subgroup, which has a significant 96% chance of securing the top rank. In Scenario 4, the data emphasize the superior predictive ability of the positive $x_{2}$ subgroup, which has a 70% probability of obtaining the highest rank. At the same time, the positive $x_{1}$ subgroup is competitive, with a 50% probability of achieving the subsequent rank. Scenario 5 illustrates the significant predictability of the $x_{2}$ positive subgroup, which boasts an impressive 96.7% probability of reaching the highest rank. Meanwhile, the $x_{1}$ positive subgroup is more likely to occupy the lowest rank, indicating its comparatively reduced predictive ability. The patterns identified in Scenario 6 closely mirror those in Scenario 5, demonstrating comparable predictive patterns in both scenarios.

Figure 1b emphasizes the importance of considering both biomarkers measured. In scenario 3, both the subgroup with both biomarker positivity and the subgroup with $x_{1}$ negativity and $x_{2}$ positivity share the first two ranks, each having a probability of 50%. Additionally, the remaining two subgroups also share the last two ranks with a probability close to 50%. In scenario 4, the subgroup that has two positive biomarkers has a 63.3% chance of being ranked first. In Scenario 5, the subgroup that has been identified with a negative $x_{1}$ and a positive $x_{2}$ biomarker showcases a 93.8% probability of securing the first position. Conversely, the subgroup that displays positive biomarkers generally receives a moderate rank, securing third place with a 76.8% probability, which indicates its fair predictive capabilities. The patterns observed in Scenario 6 are consistent with those in Scenario 5.

## 5. Discussion

In this paper, we introduce a new method for assessing experimental therapies and for identifying effective patient subsets in the field of precision medicine. The suggested Bayesian sequential design combines the advantages of the zero-inflated truncated Poisson (ZITP) regression model with the surface under the cumulative ranking curve (SUCRA) technique. The pressing need for establishing specialized techniques that identify effective patient subgroups within a precision medicine context propelled this work [42,43]. Therefore, there is an urgent need for effective methods to fully utilize the vast biomarker data available. However, biomarker information measurement in current controlled clinical trials occurs only occasionally, due to a host of factors. This inconsistency can lead to conclusions that may misrepresent the identification of truly effective subgroups. Our findings illustrate the numerous benefits of using the Bayesian sequential design in this context. In particular, the ZITP regression model is effective in managing overdispersion in count outcomes stemming from an excess of zero responses and truncation on the right end of the data, ensuring precise and reliable results.

These results reveal the intricate roles that biomarkers play in predictive and prognostic situations, particularly in terms of affecting true positive rates. As our research delved deeper into more intricate scenarios, increasing the number of measured biomarkers introduced unintended complexities. This increase inevitably resulted in smaller sample sizes for the resulting “unmissed” subsets, which runs the risk of a higher false-negative rate. Balancing precision in identifying effective subsets with the robustness of the findings is essential.

The SUCRA method expertly identifies and categorizes populations into distinct subsets based on the predictive effect. Importantly, SUCRA calculates the posterior probability of ranking by directly accounting for uncertainty. As a result, it offers a perspective that goes beyond a simple point estimate, providing a clear and intuitive classification of subgroups. This clarity is instrumental in precision medicine, where diverse patient subgroups may react differently to the same treatments.

Several other methods have been proposed to identify patient subpopulations, including decision tree algorithms, clustering, and regression-based methods. Decision tree algorithms, such as the Bayesian additive regression tree (BART), identify subgroups with differential treatment effects [44,45]. Although the BART approach can handle multiple variable types of complex models and data, it requires more computational resources and may be less easy to interpret for people unfamiliar with machine learning techniques. Clustering methods, such as *K*-means and hierarchical clustering, are used to group patients according to their similarities in covariates [46]. These methods can be useful for identifying subgroups with similar characteristics but do not provide any information on the treatment effect for each subgroup. Regression-based methods, such as logistic and linear regression, are used to model the relationship between the response variable and a set of predictor variables [38]. These methods can provide useful information about the treatment effect for each subgroup. However, they may be well suited to identifying subgroups with differential treatment effects only if the predictor variables are carefully chosen. In comparison, the SUCRA method had the advantage of being model agnostic. It may be used with any regression model to assess the treatment effect, rendering this approach versatile for identifying subgroups with differential treatment effects.

However, some limitations of the study should be noted. First, we placed ourselves in a clinical trial setting dealing with low size effects, requiring a large number of patients. This was motivated by a real clinical trial conducted in ICU patients where mild treatment effects are expected, but these could easily be adapted to larger effects in small trial samples. Second, the lack of information provided by unmeasured biomarkers impacted the results, and the proposed approach unveils its true potential when all predictive biomarkers are accounted for in the analysis. Otherwise, it is more challenging to identify subgroups with differential treatment effects, as shown by the results obtained in the simulation study. However, the current biomarker measurements in the ICU lack consistency, which may limit the ability to draw meaningful conclusions from the analysis. We placed ourselves in a Bayesian framework for estimation purposes, but this could be performed in a frequentist one. This would modify the computation of decision rules and the ranking of subsets through the SUCRA, though the P-score, its frequentist version, has been proposed [47]. These limitations highlighted the importance of having all relevant information available, including high-quality biomarker data, to achieve the most precise results when using the SUCRA method for identifying subsets.

## 6. Conclusions

In conclusion, the proposed Bayesian sequential design offers various benefits for assessing experimental treatments and identifying effective patient subgroups in precision medicine. These benefits make the method potentially useful for researchers and practitioners. However, it is critical to note that the applicability of these results to broader clinical or real-world scenarios is uncertain. Each study, including ours, has a unique context, and findings from one context may not directly apply to another. This underscores the significance of context when interpreting results and identifies potential opportunities for future research.

Further research is necessary to validate our findings and to explore broader applications. This involves creating strong and expandable techniques for handling missing or incomplete data and incorporating previous knowledge into statistical models. Given the nuanced nature of clinical scenarios and the ever-evolving landscape of precision medicine, ongoing exploration and refinement of methodologies are crucial.

## Figures and Tables

**Figure 1 jpm-13-01560-f001:**
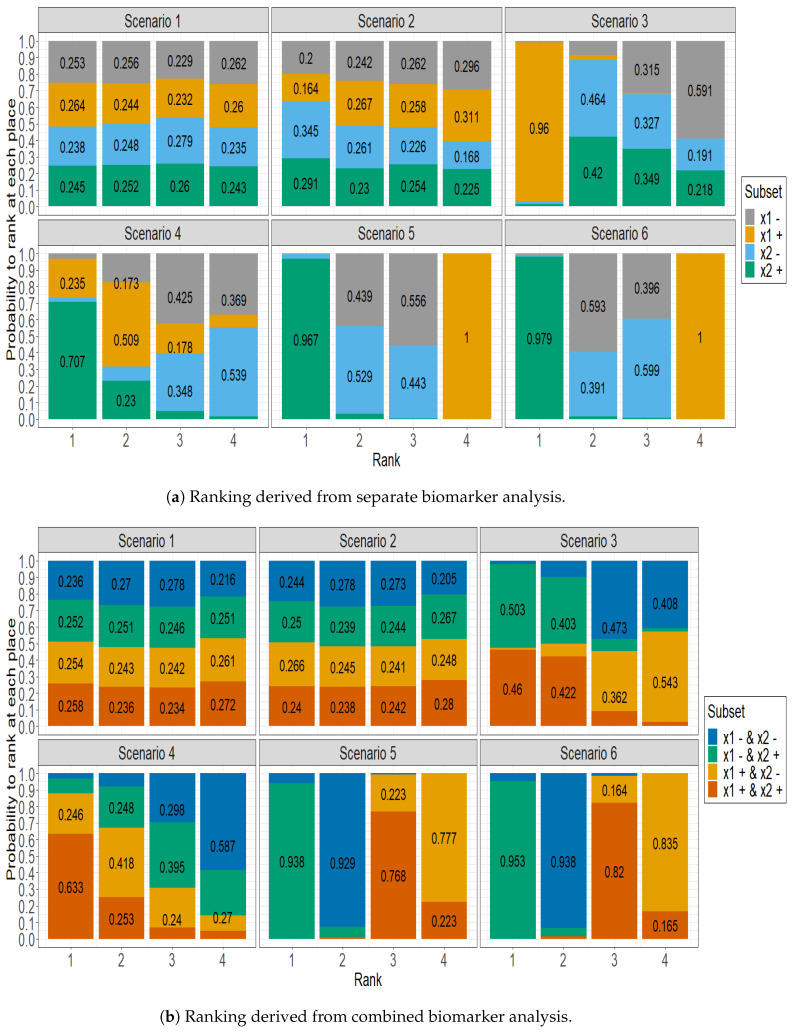
Comparative predictive rankings of patient subgroups using the SUCRA Approach. (**a**) presents the ranking of patient subgroups for each biomarker, whether or not the other biomarker was measured, based on separate modelling incorporating either biomarker. (**b**) examines the ranking of subgroups in the subsample of patients with both measured biomarkers in a complete case analysis.

**Table 1 jpm-13-01560-t001:** Simulation scenarios: True model parameters.

Scenario	$\beta_{0}$	$\beta_{1}$	$\beta_{2}$	$\beta_{T}$	$\beta_{{\mathrm{inter}}_{1}}$	$\beta_{{\mathrm{inter}}_{2}}$
1: Null	2.60	0.00	0.00	0.14	0.00	0.00
2: Prognostic effect	2.60	0.20	0.00	0.14	0.00	0.00
3: Predictive effect	2.60	0.00	0.00	0.14	0.00	0.10
4: Both prognostic and predictive effect	2.60	0.10	0.10	0.14	0.05	0.10
5: Qualitative and quantitative interactions	2.60	0.10	0.15	0.14	−0.25	0.05
6: Qualitative and quantitative interactions II	2.60	0.10	0.15	0.20	−0.25	0.05

**Table 2 jpm-13-01560-t002:** Operating characteristics of the design.

	Prevalence of Positive $x_{2}$	Average Sample Size	False/True Positive Rate	Early Stopping Rate
Scenario 1: No prognostic and predictive value of biomarker	0.2	1754	0.066	0.035
	0.4	1738	0.082	0.048
	0.5	1728	0.088	0.055
	0.6	1734	0.096	0.051
	0.8	1730	0.087	0.054
Scenario 2: Only prognostic value of biomarker $x_{1}$	0.2	1692	0.212	0.083
	0.4	1687	0.206	0.087
	0.5	1686	0.204	0.088
	0.6	1676	0.206	0.095
	0.8	1690	0.183	0.085
Scenario 3: Only predictive value of biomarker $x_{2}$	0.2	1579	0.437	0.172
	0.4	1384	0.693	0.328
	0.5	1176	0.862	0.485
	0.6	1072	0.989	0.561
	0.8	734	0.999	0.824
Scenario 4: Prognostic value of both and quantitative interaction with $x_{1}$ and $x_{2}$	0.2	1237	0.888	0.433
	0.4	932	0.994	0.668
	0.5	774	0.998	0.789
	0.6	669	0.999	0.870
	0.8	551	1.000	0.961
Scenario 5: Qualitative and quantitative interaction in both biomarker	0.2	1799	0.001	0.001
	0.4	1800	0.000	0.000
	0.5	1800	0.000	0.000
	0.6	1800	0.000	0.000
	0.8	1799	0.001	0.001
Scenario 6: Qualitative and quantitative interaction in both biomarker II	0.2	1786	0.015	0.011
	0.4	1771	0.047	0.022
	0.5	1738	0.088	0.048
	0.6	1701	0.147	0.076
	0.8	1622	0.334	0.137

## Data Availability

Not applicable.

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
