# Peer review of "Bayesian Sequential Design for Identifying and Ranking Effective Patient Subgroups in Precision Medicine in the Case of Counting Outcome Data with Inflated Zeros"

_jpm, 2023, doi:10.3390/jpm13111560_

Round 1

Reviewer 1 Report

Comments and Suggestions for Authors

The manuscript was very interesting with development of a new sequential design. However, there are some points that need to be clarified.

1.         Overall: I think analyses rather than analyzes are more common as a noun of analyze.

2.         Abstract: It is written that “demonstrated an unbiased estimation and robustness of subgroup ranking scheme”, but what does the unbiased estimation of subgroup ranking scheme mean? In addition, where are the results that evaluated unbiased estimation?

3.         Introduction: I think that an aim of clinical trial is to evaluate an effect of therapeutic intervention. Is identifying the subset of patients who respond to the experimental therapy conducted in some kinds of clinical trials? In other words, it is not certain why using SURCA is important in a clinical trial. Could you clarify about it in Introduction?

4.         2.1: Motivating example: Meaning of the number of vasopressor-free days by day 28 is unclear. Is that mean the number per month?

5.         2.2 Probability model equation (1): Is it possible that yi takes value outsize of [0,G]? If not, it is not meaningful to write “otherwise”. In addition, could you explain why ZITP model is needed instead of ZIP model?

6.         2.2 Probability model equation (3): Please write how to deal with NA values for biomarkers.

7.         2.3 & 2.4 in methods: It is not clear how subsets defined based on biomarkers are used in the analysis. How do you evaluate a superiority of therapeutic effect in each subset of biomarkers? I think that an analysis model in each subset does not correspond with the equation (3). More explanation is needed.

8.         2.2 & 2.3 & 2.4: Although incorporating a prior information by a Bayesian model is a theme of this manuscript, but it is not mentioned in 2.2 & 2.3 & 2.4. Could you clarify what is the Bayesian model you employed? In addition, how is the posterior distribution derived in 2.4?

9.         3. Simulation: Could you write an aim of simulation first? In simulation, only the performance of the proposed method was evaluated, and please write what aspects of the proposed method was evaluated in the simulation. What is the criteria of a good performance of the proposed method in the simulation.

10.     3.3. Data analysis: If non-informative prior information is used, it is not certain there is a merit in using a Bayesian model. What do you think about it? In addition, it is written that complete case analysis was used. In this case, why did you suppose NA in the subsets of biomarkers?

11.     Figure 1:In addition, I think that Figure 1 is the results of robustness or performance of ranking of the method, but it is difficult to understand what Figure 1 indicates. Could you add an explanation?

Comments on the Quality of English Language

Some parts need to be corrected.

Reviewer 2 Report

Comments and Suggestions for Authors

This is an interesting study. may I suggest that you highlight the role of noncoding RNAs in disease outcomes and in precision medicine. You can find a large section devoted to this topic in this preprint Effect of Age, Sex, and COVID-19 Vaccination History on All-Cause Mortality: Unexpected Outcomes in a Complex Biological and Social System[v1] | Preprints.org

The analysis of hospital length of stay is another application of your method. LOS has traditionally been displayed as an intiger value with any person admitted and discharged before midnight being assigned a 0 day stay.

I am not qualified to assesses the detail of your mathematical treatment. In such cases I always recommend that authors clearly state any possible hidden assumptions in the method and present a section detailing how this method has been applied in other fields and any limitations that have been raised. You have included such a section but perhaps you should expand further.

Are you able to provide any other graphical results using some real world data? maybe ask your local hospital for some length of stay data for critical care patients?

The article is well written with clear English expression.

Round 2

Reviewer 1 Report

Comments and Suggestions for Authors

Thank you for the revision. I have some additional comments.

1.         The equation (1): 0 might not be needed.

2.         If you conducted complete case analysis, is that mean that subsets with NA do not exist? In that case, it is not meaningful to suppose NA for subsets of biomarkers, isn’t it? Or if you used patients with one or more biomarker data, it isn’t called complete case analysis. It might be better to correct the word.

Comments on the Quality of English Language

It seems to be OK.
